# The Diffusion Process as a Correlation Machine: Linear Denoising Insights

**Dana Weitzner**                                                                  *danaweitzner@mail.tau.ac.il*
*Tel Aviv University*

**Mauricio Delbracio**                                                                  *mdelbra@google.com*
*Google*

**Peyman Milanfar**                                                                  *milanfar@google.com*
*Google*

**Raja Giryes**                                                                  *raja@tauex.tau.ac.il*
*Tel Aviv University*

**Reviewed on OpenReview:** *https://openreview.net/forum?id=FGDJOc27rt*

## Abstract

Recently, diffusion models have gained popularity due to their impressive generative abilities. These models learn the implicit distribution given by a training dataset, and sample new data by transforming random noise through the reverse process, which can be thought of as gradual denoising. In this work, to shed more light on the evolution of denoisers in the reverse process, we examine the generation process as a "correlation machine", where random noise is repeatedly enhanced in correlation with the implicit given distribution. To this end, we explore the linear case, where the optimal denoiser in the MSE sense is known to be the PCA projection. This enables us to connect the theory of diffusion models to the spiked covariance model, where the dependence of the denoiser on the noise level and the amount of training data can be expressed analytically, in the rank-1 case. In a series of numerical experiments, we extend this result to general low rank data, and show that low frequencies emerge earlier in the generation process, where the denoising basis vectors are more aligned to the true data with a rate depending on their eigenvalues. This model allows us to show that the linear reverse process is a generalization of the prevalent power iteration method, where the generated distribution is composed of several estimations of the given covariance, in varying stages of convergence. Finally, we empirically demonstrate the applicability of our findings beyond the linear case, in the Jacobians of deep, non-linear denoisers, used in general image generation tasks.

## 1 Introduction

Recently, diffusion models have gained much popularity as very successful generative models, showcasing impressive performance in image generation tasks (Dhariwal & Nichol, 2021; Ho et al., 2020; Song & Ermon, 2019; Song et al., 2021c). These models learn the implicit distribution given by a training dataset and sample new data by transforming random noise inputs through a reverse diffusion process, which can be thought of as gradual denoising. More formally, it has been shown in Kadkhodaie et al. (2024) that learning the underlying distribution is equivalent to optimal denoising at all noise levels.

In order to shed more light onto the mechanism behind the success of diffusion models, in this work we analyze the behavior of denoisers in the context of image generation, where pure noise is gradually processed into a sample from a given (implicit) distribution by gradual denoising. Unlike other works, e.g. Kadkhodaie

et al. (2024), we focus on the denoisers throughout the generation process, and not only on the final generated data.

To this end, we suggest the following simple model to illustrate our point. Consider the class of linear denoisers, where the optimal denoiser in MSE sense has a closed-form solution. We explore two linear denoising trajectories, corresponding to the DDPM (Ho et al., 2020) and DDIM (Song et al., 2021a) approaches of sampling. To simulate the diffusion generation process, we learn a series of projections onto noisy data at different noise levels, and use them to transform pure noise into samples from the underlying distribution. Given this simple model we can inspect the evolution of eigenvectors spanning gradual projections with decreasing noise levels, as well as the distribution of the generated data samples.

We show that the correlation of the noisy basis eigenvectors with their clean version decays as the noise level increases, with a rate determined by the eigenvalues and the size of the training dataset. In other words, we show that low frequencies, corresponding to large eigenvalues, emerge earlier in the reverse process, as empirically observed in Ho et al. (2020), and analyze how more training data contribute to generalization (Kadkhodaie et al., 2024). Analytically, this corresponds to the spiked covariance model (Johnstone, 2001), in which we bound this decay to the leading eigenvector (corresponding to the largest eigenvalue).

Next, we demonstrate the applicability of our findings to more general, non-linear deep denoisers. Although the network is not linear, its application can be written as a linear operation of the Jacobian calculated on the input image. We empirically show that the aforementioned decay of eigenvector correlations is prevalent also in the Jacobians of deep denoisers, in the final stages of image generation, thus showing the relevance of our analysis in a broader context, and not just in a simplified linear case.

## 2 Background and Related Work

Since their introduction in Sohl-Dickstein et al. (2015), diffusion models have been vastly used in image generation tasks (Dhariwal & Nichol, 2021; Ho et al., 2020; Song & Ermon, 2019; Song et al., 2021c), more general computer vision tasks (Amit et al., 2021; Baranchuk et al., 2022; Brempong et al., 2022; Cai et al., 2020), and in other domains such as natural language processing (Austin et al., 2021; Hoogeboom et al., 2021; Li et al., 2022; Savinov et al., 2022; Yu et al., 2022) and temporal data modeling (Alcaraz & Strodthoff, 2023; Chen et al., 2021; Kong et al., 2021; Rasul et al., 2021; Tashiro et al., 2021). On top of their practical success, different flavors of training and sampling have risen based on interesting theoretical reasoning, e.g., considering the statistical properties of the intermediate data (Song et al., 2021a; Sohl-Dickstein et al., 2015), or by framing the problem in the form of stochastic differential equations (SDEs) (Karras et al., 2022; Song et al., 2021b;c; Chen et al., 2024) or score based generative models (Song & Ermon, 2019; 2020). In this work, we look at diffusion models in the context of iterative denoising, and focus on the properties of the learned denoiser (Milanfar & Delbracio, 2024).

Recently, the work in Kadkhodaie et al. (2024) showed that the learned denoising functions are equivalent to a shrinkage operation in a basis adapted to the underlying image. In this sense, the diffusion denoiser is an adaptive filter (Milanfar, 2013; Talebi & Milanfar, 2014; 2016). While they focus on the analysis of the nonlinear denoiser at the point of the final generated data, we are interested in the evolution (adaptation) of the denoiser throughout the generation process, and its dependence on the noise level. To this end, we suggest a simple linear denoising model, presented in Section 3. In this case, the (optimal) denoiser does not depend on the underlying image, and its dependence on the noise level can be traced analytically, as we show hereafter.

Due to their phenomenal empirical success, some attempts have been devoted towards providing theory supporting the sample and iteration complexity of diffusion models. The current body of work can be generally parted to attaining iteration complexity bounds assuming approximately accurate scores (Li et al., 2024b;a; Chen et al., 2023b; Huang et al., 2024; Benton et al., 2024), and to assessing the sample complexity to learn the score functions (Chen et al., 2023a; Block et al., 2020; Biroli & Mézard, 2023). Among these works, many assume a low dimensional data distribution (Bortoli, 2022; Li & Yan, 2024; Oko et al., 2023; Chen et al., 2023a; Wang et al., 2024), which is a reasonable assumption in practice (see e.g., Pope et al. (2021)). Yet, it might particularly explain the gap between the current iteration bounds and the much lower

complexity apparent in practice (Li & Yan, 2024). In our work, we consider linear models and deduce a linear sample complexity bound associated with learning the score function in Sec. 4 and discuss the trade-offs of the synthesis conversion rate in Sec. 4.1. The previous works mentioned above mainly develop bounds assuming specific samplers and scaling details, which differ from our setting. In addition, they generally bound the Total Variation distance (under varying assumptions on the target distributions), which is not trivial to translate to the generated covariance matrix that we focus on even in the linear Gaussian case (Devroye et al., 2018). The difference in our setting enables us to connect the theory of diffusion models to a broad body of work concerning the spiked covariance model (Johnstone, 2001), and supports the analysis of denoising diffusion as a correlation machine, which is the main purpose of this paper.

In the setting of Statistical Mechanics, the work in Biroli & Mézard (2023) analyzes diffusion models in very large dimensions, focusing on the Curie-Weiss model of ferromagnetism. As an introduction to their work, they also discuss a simple linear score model, in the context of the sample complexity of learning the score function. They focus their discussion on the case of Gaussian data, where the eigenvalues of the covariance matrices can be typically characterized. Relatedly, the work in Wang & Vastola (2024) recently showed that the learned neural score is dominated by its Gaussian approximation for moderate to high noise scales, and supply both theoretical and empirical arguments to support this claim. Compared to these works, we consider data that reside in a low dimensional subspace, with no specific distribution, described in Sec. 4. We limit the denoiser to be linear and focus on two stochastic sampling trajectories, which give rise to the spiked covariance model.

Power iteration is a fundamental algorithm for approximating the dominant eigenvalue and eigenvector of a matrix. It relies on iteratively multiplying an initial vector by the matrix, where its convergence rate is proportional to the ratio of the largest and second-largest eigenvalues. The method's simplicity and scalability have made it a cornerstone in various fields, including numerical linear algebra, machine learning, and graph theory. For the ease of reading, we include a formal presentation of the method and discuss its convergence in Appendix A. In this work, we shall show how a linear denoising chain converges in mean to the celebrated power iteration method.

## 3 Linear Diffusion - Problem Setup

For our analysis, we define the following simple iterative linear generation model. First, define the standard diffusion model. Let $q_D$ denote the natural data distribution and let $x_0 \sim q_D$ be a sample from the natural data ($x_0 \in \mathbb{R}^d$). The forward (diffusion) process is defined (Ho et al., 2020) by

$$q(x_t|x_{t-1}) = \mathcal{N}(\sqrt{1-\beta_t}x_{t-1}, \beta_t \mathbf{I}) \tag{1}$$

for some fixed noise schedule $\{\beta_t\}_{t=1}^T$ and $x_0 \sim q_D$. It can be shown that

$$q(x_t|x_0) = \mathcal{N}(\sqrt{\bar{\alpha}_t}x_0, (1-\bar{\alpha}_t)\mathbf{I}), \tag{2}$$

where $\alpha_t = 1 - \beta_t$ and $\bar{\alpha}_t = \Pi_{s=1}^t \alpha_s$. The reverse (generation) process is defined using a parameterized distribution model $p_\theta$, generally defined by the Markov process

$$p_\theta(x_{0:T}) = p(x_T)\Pi_{t=1}^T p_\theta(x_{t-1}|x_t), \tag{3}$$

$$p_\theta(x_{t-1}|x_t) \triangleq \mathcal{N}(\mu_\theta(x_t, t), \Sigma_\theta(x_t, t)), \tag{4}$$

where $p(x_T) = \mathcal{N}(0, \mathbf{I})$. By choices of parametrization and loss manipulations (see (Ho et al., 2020)), one generally learns to estimate the error $\epsilon_\theta(x_t, t)$, where

$$\mu_\theta(x_t, t) = \frac{1}{\sqrt{\alpha_t}}\left(x_t - \frac{\beta_t}{\sqrt{1-\bar{\alpha}_t}}\epsilon_\theta(x_t, t)\right), \tag{5}$$

$\Sigma_\theta(x_t, t) = e_t^2 \mathbf{I}$, and $e_t$ is a designed schedule. Thus, the reverse process can be expressed as a denoising chain

$$D_t(x_t) = \frac{1}{\sqrt{\alpha_t}}\left(x_t - \frac{\beta_t}{\sqrt{1-\bar{\alpha}_t}}\epsilon_\theta(x_t, t)\right) + e_t z, \tag{6}$$

where $z \sim \mathcal{N}(0, \mathbf{I})$ and $z_1 = 0$. This is a stochastic denoiser which preserves the Markovian property of the forward process. Later versions suggested similar (non-Markovian) deterministic denoisers, e.g., DDIM (Song et al., 2021a), or more general stochastic denoiser chains, for a continuous forward model (InDI (Delbracio & Milanfar, 2023)).

Now, we turn to define our linear setting within the diffusion context. For our simplified model, consider the process (without scaling),

$$q(x_t|x_{t-1}) = \mathcal{N}(x_{t-1}, \sigma_t^2 \mathbf{I}). \tag{7}$$

This implies that $x_t = x_{t-1} + \epsilon_{\sigma_t}$, where $\epsilon_{\sigma_t} \sim \mathcal{N}(0, \sigma_t^2 \mathbf{I})$ for some fixed noise schedule $\{\sigma_t\}_{t=1}^T$. We discard the scaling to comply with previous analysis of the spiked covariance model (Nadler, 2008) (more details in Section 4). This corresponds to the "Exploding Variance" formulation, used with Langevin dynamics to sample data as a variant of score based diffusion models (Song & Ermon, 2019; Song et al., 2021c; Song & Ermon, 2020). We choose to present the "standard" diffusion models in the setting of denoising diffusion (Ho et al., 2020) and not using the score-based approach entirely, as we focus our discussion on the qualities of the denoiser. In our case, we restrict the denoisers to be a linear function of $x_t$. For the reverse process, we shall now define two linear denoising trajectories, corresponding to different approaches of diffusion models.

Recall that at each time step $x_t = x_{t-1} + \epsilon_{\sigma_t}$, where $\epsilon_{\sigma_t} \sim \mathcal{N}(0, \sigma_t^2 \mathbf{I})$ (Equation 7). Since the noise is assumed to be Gaussian, we can write $x_t = x_0 + \epsilon_{\bar{\sigma}_t}$, where

$$\bar{\sigma}_t = \sqrt{\sum_{i=0}^{t} \sigma_i^2}, \tag{8}$$

where $\epsilon_{\bar{\sigma}_t} \sim \mathcal{N}(0, \bar{\sigma}_t^2 \mathbf{I})$, i.e., $\bar{\sigma}_t$ is the accumulated noise at time $t$. Let $\Sigma_t$ be the covariance of the noisy data at time $t$,

$$\Sigma_t = \mathbb{E}x_0 x_0^\dagger + \bar{\sigma}_t^2 \mathbf{I} \triangleq \Sigma_0 + \bar{\sigma}_t^2 \mathbf{I}. \tag{9}$$

Considering an intermediate denoising step from $t+1 \to t$, the optimal linear denoiser $D_{t+1\to t}^*$ in the $\ell_2$ sense is the minimizer of the loss

$$\ell_{t+1\to t} = \mathbb{E}_{x_t, \epsilon_{\sigma_{t+1}}} \|D_{t+1\to t}(x_t + \epsilon_{\sigma_t+1}) - x_t\|_2^2, \tag{10}$$

given by

$$D_{t+1\to t}^* = (\Sigma_t + \sigma_{t+1}^2 \mathbb{I})^{-1} \Sigma_t \tag{11}$$

(full derivation in Appendix D). Notice, that in the limit of diminishing $\sigma_t$,

$$D_{t+1\to t}^* = U_t \begin{pmatrix} \frac{\lambda_0}{\lambda_0 + \sigma_{t+1}^2} & & \\ & \ddots & \\ & & \frac{\lambda_{r-1}}{\lambda_{r-1} + \sigma_{t+1}^2} \end{pmatrix} U_t^\dagger \underset{\sigma_{t+1}\to 0}{\to} U_t U_t^\dagger \triangleq D_{\mathrm{PCA}}^t, \tag{12}$$

where $U_t$ is the diagonalizing basis of $\Sigma_t$ with the time dependent spectrum $\{\lambda_i\}$, where we omit an explicit temporal dependency in the notation for brevity (more on that in Sec. 4). Our sampling process is based on the sequential application of $D_{\mathrm{PCA}}^t$, which is the projection on perturbed principal components with respect to the clean data distribution, followed by the addition of noise with variance $\sigma_t$. This generation path is in the spirit of (Ho et al., 2020), with its gradual temporal denoising. However, $D_{PCA}^t$ is a deterministic denoiser given the sampling of training data and noise, which does not depend either on $x_t$ or on $x_0$. In Section 4, we analyze the change in the finite sample approximation of $D_{PCA}^t$ over time, to study its evolution along the generation trajectory.

Given a similar $\ell_2$ loss (to Equation 10), an alternative denoising chain can use multiple estimations of $x_0$, in the essence of Song et al. (2021a). The corresponding loss is thus

$$\ell_{t\to 0} = \mathbb{E}_{x_t, \epsilon_{\bar{\sigma}_t}} \|D_{t\to 0}(x_0 + \epsilon_{\bar{\sigma}_t}) - x_0\|_2^2, \tag{13}$$

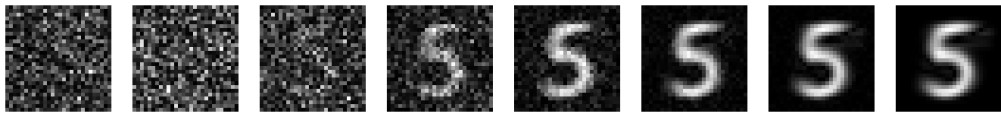

Figure 1: Digit generation from pure noise (class conditioned). The reverse process runs from left to right.

where $\bar{\sigma}_t$ is the overall added noise (defined in Equation 8). The adequate denoising chain in this case is the application of $D^*_{t\to0}$ to estimate $x_0$, followed by the addition of noise with the appropriate variance $\bar{\sigma}^2_{t-1}$, before the iterative application of $D^*_{t-1\to0}$. In this case, the optimal denoiser is given by

$$D^*_{t\to0} = (\Sigma_0 + \bar{\sigma}^2_t \mathbb{I})^{-1}\Sigma_0. \tag{14}$$

Despite the different approaches the two paths represent, their resulting denoising chains exhibit similar properties - in both cases, the appearance of frequencies in the generated images is gradual, where low frequencies are first to emerge.

**Empirical Demonstration of a Linear Diffusion Model.** To illustrate the forward and backward processes in the linear case, we perform a numerical simulation using the MNIST dataset, which is simple enough to be estimated via a linear model. We start here with the training and generation procedures, and use the same setting and trained denoisers to demonstrate our findings throughout the paper.

In the following experiment we simulate the process described above using the MNIST dataset (we use the default train / test splits). In the class conditioned case, we learn a (finite sample approximation of a) PCA denoiser with 30 components for each time step where $x_t = x_{t-1} + \epsilon_{\sigma_t}$, $\sigma_t \propto t$, and $T = 65$ iterations. Figure 1 shows a (decimated) example of digit generation from pure noise, where we apply the sequence of (finite sample approximation of) denoisers $D^t_{PCA}$. In order to understand the reverse process, we now turn to analyze the gradual change in the denoisers, that might be expressed by the angle between the clean and noisy components over time.

**Notations.** We use $A_t$ to denote the matrix $A$ at time $t$, and $a^t_i$ to denote the $i$th column of $A_t$.

## 4 Linear Diffusion as Basis Perturbation

We now turn to analyze the linear model presented above and show how the generation process can be seen as a kernel "correlation machine". Specifically, we are interested in the temporal (i.e., noise level) dependence of the finite sample approximation of $D^t_{PCA}$ throughout the generation process. Assume that the data distribution is such that its population covariance is given by

$$\Sigma_0 = \mathbb{E}x_0 x_0^\dagger = \sum_{i=0}^{r-1} \lambda_i^2 u_i u_i^\dagger, \tag{15}$$

where $r-1 < d$, i.e., the data reside in a low dimensional subspace (which is generally true for natural data). Thus, the population covariance at time $t$ is given by

$$\Sigma_t = \sum_{i=0}^{r-1} \lambda_i^2 u_i u_i^\dagger + \bar{\sigma}^2_t \mathbf{I}. \tag{16}$$

This data model is known as the "spiked model" (Johnstone, 2001), with a vast body of work covering the distribution and identifiability of the spikes spectrum (e.g., (Nadler, 2008); for more information, please see Appendix B). Throughout the paper, we use the term "index" to refer to to the index $i$ in Equation 16, where the eigenvalues $\lambda_i$ are ordered largest to smallest.

Given $n$ samples concatenated as columns in the matrix $X_0$ and $n$ noise vectors in the columns of $E_{\bar{\sigma}_t}$, at each time step we learn the PCA basis associated with $X_t = X_0 + E_{\bar{\sigma}_t}$, by the diagonalization of the sample covariate matrix

$$\hat{\Sigma}_t = \frac{1}{n}X_t X_t^\dagger = \frac{1}{n}(X_0 X_0^\dagger + X_0 E_{\bar{\sigma}_t}^\dagger + E_{\bar{\sigma}_t} X_0^\dagger + E_{\bar{\sigma}_t} E_{\bar{\sigma}_t}^\dagger) \triangleq \hat{U}_t \hat{S}_t \hat{U}_t^\dagger. \tag{17}$$

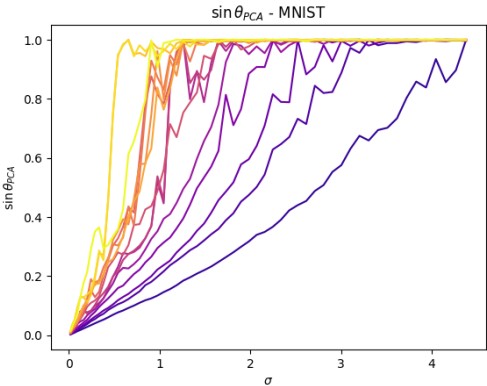

Figure 2: The sine of the angle between the clean principal components and their noisy versions, colored by the order of the eigenvalues (the darkest being largest eigenvalue). Low frequencies emerge earlier in the generation process (at higher noise levels). This motivates Assumption 4.1, that extends Equation 19 to higher ranks.

Thus, during the reverse process, at each time step we apply the projection

$$P_t \triangleq \hat{U}_t \hat{U}_t^\dagger. \tag{18}$$

In order to understand the generation process, we analyze the decay of the product $\langle \hat{u}_i^t, u_i \rangle$ over time, where $\hat{u}_i^t$ is the $i$th column of $\hat{U}_t$. Note, that there are two drivers of change in the perturbation of $u_i$ to $\hat{u}_i^t$. The first being the added noise, i.e., $\|\Sigma_t - \Sigma_0\|$. This is the key in the diffusion process and our main focus. The second, is in the finite sample approximation $\left\|\hat{\Sigma}_t - \Sigma_t\right\|$. This source of error is interesting in the context of sample complexity, as it encompasses the approximation of the denoiser learned from a finite dataset, the equivalent of the sample complexity of learning the score function (Chen et al., 2023a; Block et al., 2020; Biroli & Mézard, 2023). For the rank-1 case, Nadler (2008) presented a finite sample theorem which holds with high probability for the closeness between the leading eigenvalue and eigenvector of sample and population PCA under a spiked covariance model similar to Equation 16. They bound the angle between the leading empirical eigenvector and its population counterpart with approximately $\mathcal{O}(d)$ sample complexity, and a linear dependence on the noise level. Their bound can be approximately expressed by

$$\mathbb{E}\sin\theta_{\mathrm{PCA}} = \mathbb{E}\sqrt{1 - \langle \hat{u}^t, u \rangle^2} \approx \frac{\bar{\sigma}_t}{\lambda}\sqrt{\frac{d}{n}}, \tag{19}$$

where $\bar{\sigma}_t$ is assumed to be small and $d \gg 1$ (for the full derivation, please see Appendix E). This result shows that the leading eigenvector rotates at a rate proportional to the noise level. Our experiments on the MNIST dataset (detailed in Section 4.1) show that this is a good approximation in practice, also for the rank-r case (Fig. 2). We chose to only consider the leading part of the bound in Nadler (2008), as it captures its essence while significantly simplifying the writing. This approximation is justified since it was also shown to be sharp (Nadler, 2008, Corollary 1), and empirically by our simulations (and specifically by Fig. 2).

Notice that in Equation 19 the angle is inversely linked to the eigenvalue, inferring a slower change with higher eigenvalues. In the reverse process, we gradually move from pure noise or high noise levels to smaller noise variance. Given the lower slope of the components corresponding to larger eigenvalues, we interpret the result in Fig. 2 as the earlier emergence of low frequencies in the generation process. The first component to be visible in the generated image is the one with the largest eigenvector, as it is the first one that shows a correlation in high noise levels. Throughout the generation process, when the noise level decreases, the next components take presence, by the order of their associated eigenvalue - from the larger to the smaller. Finally, the components with the smallest eigenvalues appear when the noise level is low. In other words, we show that low frequencies, corresponding to large eigenvalues, emerge earlier in the reverse process, as empirically observed in Ho et al. (2020). This phenomenon resonates with recent findings in the context of

exposure bias (Li et al., 2024c; Ning et al., 2023), who describe the diffusion process as inherently two-staged: initially pushing samples toward the data manifold, then committing to individual modes. For more context regarding the connection between the principal component index and frequency, please see Appendix C.

In the linear case, Equation 19 shows that the diffusion model's sample complexity is determined by the sample complexity of PCA, with a linear dependence on the dimension of the data. To further enhance our understanding of the relationship between the amount of training data and generalization, we repeat the experiment with varying datasets sizes. Figure 3 shows the angle to noise profile for selected principal components, with the indices $0, 5, 10$ (left to right; index 0 corresponds to the largest in a list of ordered eigenvalues, that was covered by the result in Nadler (2008)). Increasing the amount of training data improves robustness to noise and enables the emergence of higher frequency components at higher noise levels, thereby capturing more nuances in the generated data. Thus, the two right plots in Fig. 3 show that the data behavior in Equation 19 is a good approximation also in higher indices.

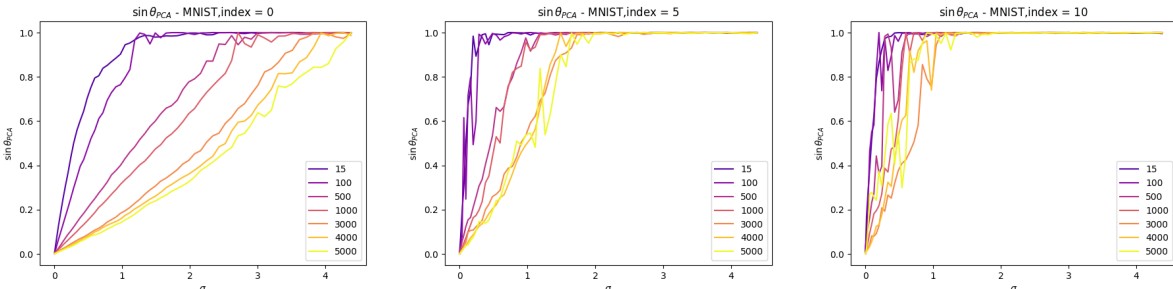

Figure 3: Effect of dataset size. The plots show $\sin \theta_{\mathrm{PCA}}$ at different noise levels when trained on datasets with increasing size (lighter color). Each plot is of a different component index, for indices $0, 5, 10$ (left to right; index 0 corresponds to the largest eigenvalue). Increasing the amount of training data improves the robustness to noise, and allows the appearance of high frequencies at higher noise levels, hence capturing more data nuances in the generated data and better generalization.

### 4.1 The Generated Distribution

We now turn to discuss the distribution of the generated output, and how it relates to the natural data distribution. We shall start from the first sampling path, considering the PCA based denoising (Equation 18), and then describe the second generation trajectory, using estimations of $x_0$ (Equation 14).

Given our linear model, the generated output is given by

$$\hat{x} = \Sigma_{t=0}^{T} \Pi_{\tau=0}^{t} P_\tau \xi_t = P_0 \cdots P_T \xi_T + \cdots + P_0 \xi_0 \tag{20}$$

for $\xi_t \sim \mathcal{N}(0, \sigma_t)$. For the ease of writing, define $\mathcal{P}_t = \Pi_{\tau=0}^{t} P_\tau$, and so $\hat{x} = \Sigma_{t=0}^{T} \mathcal{P}_t \xi_t$.

Other than the visual aesthetic of the generated images, we are interested in their distribution, and how well it represents the natural distribution of training images. Thus, we would like to compare the generated covariance $\mathbb{E}\hat{x}\hat{x}^\dagger$ to the natural covariance $\Sigma_0$. We start our analysis by focusing on the first summand comprising $\hat{x}$,

$$\hat{x}_T = \mathcal{P}_T \xi_T. \tag{21}$$

In this context, a natural comparison is the power iteration (PI) method, which may be used to estimate the leading eigenvector of a matrix. This can be seen as another iterative form of generating data from random vectors. Unlike our projection, in PI we "project" a random vector onto the entire matrix, i.e. including the eigenvalues. In this case the denoiser would be $D_{PI}^t = \Sigma_0 \forall t$, where we ignore the normalization and focus on the direction of the final vector, since there is no normalization constraint for generated data in diffusion models.

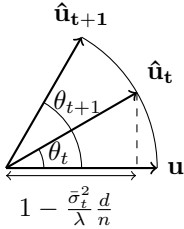

Figure 4: Schematic illustration of the basis perturbation, per index.

We now turn to show how the reverse process performed by a repeated denoising as in Equation 21 converges in mean to PI. To this end, we make the following assumptions, that are expansions of the result 19 from Nadler (2008) to the rank-r case. The first describes the correlations of the same index with its noisy versions, and the second describes the cross index correlations (that are relevant due to the noise in the finite sample).

**Assumption 4.1.** *Assume that Equation 19 holds for all eigenvectors, i.e.,*

$$\mathbb{E}\sqrt{1 - \langle \hat{u}_i^t, u_i \rangle^2} \approx \frac{\bar{\sigma}_t}{\lambda_i}\sqrt{\frac{d}{n}}, \tag{22}$$

*for $i = 0, \ldots, r-1$.*

This assumption is the extension of Equation 19 to higher ranks, and is motivated by our simulations (Fig. 2). In addition, we make the following assumption regarding the cross products of components of different indices, at consecutive time steps.

**Assumption 4.2.** *For each index $i$ there exists a time $\tau_i$, where for $t \leq \tau_i$ and $j \leq i$,*

$$\mathbb{E}\langle \hat{u}_i^t, \hat{u}_j^{t+1} \rangle = 0. \tag{23}$$

*In addition, $\tau_i > \tau_j$ for $i < j$.*

This assumption is a supported by our simulations in Fig. 5, and will be further discussed hereafter. Assumptions 4.1, 4.2, are an extension of Nadler (2008) to higher ranks. We leave their explicit derivation to future work, and focus on their implications to linear diffusion.

We are now ready to state our main result.

**Theorem 4.3** (Convergence to Power Iteration). *Let $\sigma_t = \frac{1}{T}$, $t = 0, \ldots, T$. Assuming 4.1, 4.2, in the limit $T \to \infty$,*

$$\mathbb{E}\hat{x}_T \hat{x}_T^\dagger \propto \hat{u}_0 \hat{u}_0^\dagger, \tag{24}$$

*where $\hat{x}_T = \mathcal{P}_T \xi_T$, as defined in Equation 21.*

*Proof.* Let us analyze the product in Equation 21 to show how it relates to the power method. The linear operator representing the reverse process can be written as

$$\mathcal{P}_T = \hat{U}_0 \Pi_{t=0}^{T-1} (\hat{U}_t^\dagger \hat{U}_{t+1}) \hat{U}_T^\dagger. \tag{25}$$

The matrix product $\hat{U}_t^\dagger \hat{U}_{t+1}$ can be analyzed using the extension of Equation 19 to higher ranks. Given 4.1, the expected inner product with the natural population data component $u_i$ is given by

$$\mathbb{E}\langle \hat{u}_i^t, u_i \rangle \approx 1 - \frac{\bar{\sigma}_t^2}{\lambda_i^2}\frac{d}{n}. \tag{26}$$

The evolution of this product over time is depicted in Figure 4. We are interested in the projection of $\hat{u}_{t+1}$ onto $\hat{u}_t$, which is the cosine of the angle $\Delta\theta = \theta_{t+1} - \theta_t$. This angle is tractable for small noise levels, so we

divide our analysis to two parts: $0 \le t \le \tau$ and $\tau \le t \le T$, where the choice of $\tau$ will soon be motivated. In addition, we assume adequate sample complexity, such that $\mathbb{E}\langle \hat{u}_i^t, u_i^t \rangle \approx 0$.

First, we inspect the limit of $t \to 0$ ($0 \le t \le \tau$). For small angles, we can write

$$\Delta\theta = \arccos\left(1 - \frac{\bar{\sigma}_{t+1}^2}{\lambda^2}\frac{d}{n}\right) - \arccos\left(1 - \frac{\bar{\sigma}_t^2}{\lambda^2}\frac{d}{n}\right) \approx \frac{d}{\lambda^2 n}(\bar{\sigma}_{t+1}^2 - \bar{\sigma}_t^2) = \frac{\sigma_{t+1}^2 d}{\lambda^2 n}, \tag{27}$$

since $\arccos\theta \approx \frac{\pi}{2} - \theta$ and $\bar{\sigma}_t^2 = \sum_{\tau=0}^t \sigma_\tau^2$. The diagonal elements in $U_t^\dagger U_{t+1}$ are then given by

$$\mathbb{E}\langle \hat{u}_i^t, \hat{u}_i^{t+1} \rangle \approx \cos\frac{\sigma_{t+1}^2 d}{\lambda_i^2 n}, \tag{28}$$

where the off-diagonal elements are negligible, since

$$\mathbb{E}\langle \hat{u}_i^t, \hat{u}_j^{t+1} \rangle \approx \mathbb{E}\langle \hat{u}_i^t, \hat{u}_j^t \rangle = 0, \tag{29}$$

which holds for $t \le \tau_{r-1}$ by Assumption 4.2. Notice, that in small angles, $\langle \hat{u}_i^t - \hat{u}_i^{t+1}, u \rangle = (\sigma_{t+1}^2 d)/(\lambda^2 n) \to 0$, so the vectors $\hat{u}_i^t$ are co planar, as depicted in Figure 4. Thus, the time point basis correlations $\hat{U}_t^\dagger \hat{U}_{t+1}$ form an approximately diagonal matrix with the fraction $c_i \triangleq \cos\frac{\sigma_{t+1}^2 d}{\lambda_i^2 n}$ on the diagonal, where $c_i > c_j$ for $i < j$. We eliminate the dependence of $c_i$ on $t$ by choosing the constant schedule $\sigma_t = 1/T$ $\forall t$, to simplify the proof. However, many schedules can be used, as long as $c_{i,t} > c_{j,t}$ remains correct. Define the partial linear diffusion operator until time $\tau$ by $\mathbb{E}\mathcal{P}_\tau = \Pi_{t=0}^\tau P_t$. Then

$$\mathbb{E}\mathcal{P}_\tau = \hat{U}_0 \begin{pmatrix} c_0^\tau & & \\ & \ddots & \\ & & c_{r-1}^\tau \end{pmatrix} \hat{U}_\tau^\dagger = \hat{U}_0 c_0^\tau \begin{pmatrix} 1 & & \\ & \left(\frac{c_1}{c_0}\right)^\tau & \\ & & \ddots \end{pmatrix} \hat{U}_\tau^\dagger \xrightarrow[\tau \to T]{} \hat{U}_0 \begin{pmatrix} c_0^\tau & 0 & \\ & & \\ & & \ddots \end{pmatrix} \hat{U}_\tau^\dagger, \tag{30}$$

where the diagonal elements decay as $\tau$ grows larger, since $c_i > c_j$ for $i < j$. Similarly to power iteration, the convergence rate depends on the ratio $c_1/c_0$. The convergence rate might not be fast enough for the process to converge while the small angles approximation still holds. Thus, we continue with the second phase of our analysis, showing the convergence of the full reverse process.

We now turn to analyze the phase where $\tau \le t \le T$. In high noise levels, the correlation with the natural basis is low, and the products $\hat{U}_t^\dagger \hat{U}_{t+1}$ are not exactly diagonal. However, the correlation "leaks" to a close neighborhood of the original component and the temporal products are still somewhat concentrated around their diagonal. This process happens in accordance with Equation 4.1, where the spreading of the diagonal elements happens for high indices in lower values of $t$ (less noise is needed to spread the correlation). This leads us to Assumption 4.2, claiming that for each index $i$ there exists a time $\tau_i$ after which the small angle approximation does not hold; $\tau_i > \tau_j$ for $i < j$. This is apparent in practice, and depicted in 5 (left image per duo). However, given the decaying diagonal structure of the partial operator $\mathcal{P}_\tau$, we will now show that 4.2 is sufficient for the total operator to converge as desired.

Suppose we added one more matrix multiplication to our former analysis, i.e. observe

$$\mathbb{E}\mathcal{P}_\tau \hat{U}_{\tau+1} = \hat{U}_0 c_0^\tau \begin{pmatrix} 1 & & \\ & \left(\frac{c_1}{c_0}\right)^\tau & \\ & & \ddots \end{pmatrix} \hat{U}_\tau^\dagger \hat{U}_{\tau+1}. \tag{31}$$

Assumption 4.2 guarantees $\hat{U}_\tau^\dagger \hat{U}_{\tau+1}$ is diagonal just enough not to spoil the diagonality of the next partial operator $\mathbb{E}\mathcal{P}_{\tau+1}$. To see this, let us inspect some intermediate index $i$, where the entries in $j > i$ are already practically zero. Thus, we have

$$\mathbb{E}\mathcal{P}_{\tau_i}\hat{U}_{\tau_i+1} = \hat{U}_0 c_0^{\tau_i+1} \underbrace{\begin{pmatrix} 1 & & & \\ & \ddots & & \\ & & \left(\frac{c_i}{c_0}\right)^{\tau_i} & \\ & & & \mathbb{O} \end{pmatrix}}_{\triangleq C_{\tau_i}} \begin{pmatrix} 1 & & & \\ & \ddots & & \\ & & \frac{c_i}{c_0} & \\ & & & \mathbb{A} \end{pmatrix} = \hat{U}_0 c_0^{\tau_i+1} \begin{pmatrix} 1 & & & \\ & \ddots & & \\ & & \left(\frac{c_i}{c_0}\right)^{\tau_i+1} & \\ & & & \mathbb{O} \end{pmatrix}$$



Figure 5: The time point basis correlation matrices $\hat{U}_\tau^\dagger \hat{U}_{\tau+1}$ (left per pair), together with the partial product $\Pi_{t=0}^\tau(\hat{U}_t^\dagger \hat{U}_{t+1})$ (right per pair) at different time points. This justifies Assumption 4.2, and shows that the total projection (bottom right image, for $\tau = T$) converges to the first eigenvector, similarly to the power method.

where $\mathbb{O}$ is a block of zeros and $\mathbb{A}$ is a block matrix the same size as $\mathbb{O}$, that can have nonzero entries, by Assumption 4.2. Since the elements of the partial product $C_{\tau_i}$ decay faster with $i$ than any single product $\hat{U}_{\tau_i}^\dagger \hat{U}_{\tau_i+1}$, $C_{\tau_i+1}$ is also diagonal. Overall, the final product is a diagonal matrix with a spectrum that converges to be concentrated around the first eigenvalue, where we can control the distribution of the generated data by the choice of the diffusion parameters. Figure 5 shows our simulation of the process, where $\hat{U}_\tau^\dagger \hat{U}_{\tau+1}$ (the left in each duo) is approximately diagonal until a certain index, and the total product (right in each duo) is also approximately diagonal (with less indices than its counterpart). This supports both assumption 4.2 and the result stated by this theorem. $\qquad\square$

Thus, the generated output is a combination of a (purely) noisy image that was repeatedly correlated to converge to $v_0$ (as shown above), with generally lower noise levels that are "lightly" correlated, although to the cleaner projection operators. The generated output can thus be seen as a combination of three conceptual parts, with a different balance of the noise level and the portrayed components.

**The first eigenvector**    The first part of the sum in Equation 20 is $P_0 \cdots P_T \xi_T$, the estimation of the eigenvector with the largest eigenvalue, as shown above theoretically in Equation 30 and empirically in the rightmost matrix in Fig. 5. The "strongest" noise is repeatedly correlated to be concentrated around the first eigenvector.

**The entire (clean) spectrum**    The last part in Equation 20 is $P_0 \xi_0$, a weak noise level that is spread across all components. This noise is very lightly and not repeatedly correlated, although to a clean version of the natural data basis.

**In between**    The third part consists of all the intermediate products $\Pi_{\tau=0}^t P_\tau \xi_t$. The product operators $\Pi_{\tau=0}^t P_\tau$ preserve varying parts of the natural spectrum, according to $t$ - as $t$ grows, the total projection tends to retain only the components associated with larger eigenvalues. This can be seen in Fig. 5. The right matrix in each pair shows the product $\Pi_{\tau=0}^t P_\tau$ for varying values of $t$. The total projections range from the entire spectrum (left) to only the leading eigenvalue (right). In between, the products are diagonal matrices where the entries in the indices of the smaller eigenvalues have already diminished, in a similar way to the convergence described in Equation 30.

Thus, we get a combination of a solid estimation of the leading eigenvector, together with a more uniform and week sampling of the components with low eigenvalues in the natural data basis. In between, the intermediate projections are at different levels of convergence to the leading eigenvector, hence tend to be more concentrated on components with large eigenvalues as $t \to T$. The freedom in choice of schedule $\{\xi_t\}_{t=0}^T$, allows control of the spread of the final distribution on the natural data components.

The sample trajectory we consider in Equation 20 is stochastic. However, our analysis also covers a deterministic sampling path, where the denoisers are repeatedly applied, with no added noise. This deterministic sampling is in fact portrayed by the first summand in the generated output Equation 20, which we show to

converge in mean to the first eigenvector, similarly to the power iteration. Under the limitation of the linear model, this also implies that this deterministic sampling is insufficient to capture the training data.

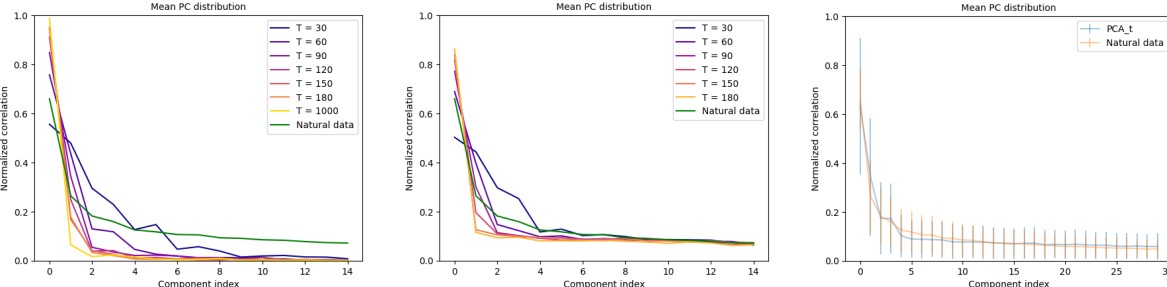

Figure 6: The empirical distribution of generated images over the natural principal components, with (middle) and without (left) injected noise. On the right - the best configuration with the generated standard deviation (see Sec. 4.1).

To inspect this, we plot the empirical distribution of generated images over the clean PCs, given by

$$p_i = \frac{1}{n} \sum_{j=1}^{n} \frac{|\langle \hat{u}_i, \hat{x}^j \rangle|}{\|\hat{x}^j\|_2}, \tag{32}$$

where $\hat{u}_i$ is the clean principal component with index $i$ (defined in Equation 16) and $\hat{x}^j$ is a generated sample, out of $n$ examples. Figure 6 shows the empirical distribution of generated images over the clean principal components. On the left, we plot the distribution without injected noise (i.e., $\hat{x} = \mathcal{P}_T \xi$), for various values of $T$. As we show above, the distribution tends to be concentrated on the first eigenvector as $T$ increases. The center plot shows the distribution of the process including the injected noise in the intermediate denoising steps. While in the low indices the dominant behavior is similar to the former case, the higher orders do not converge to zero and maintain their presence in the generated distribution. We note, that more sophisticated nonlinear deterministic samplers might not require the injection of noise in order to converge to the natural data distribution (e.g. (Lu et al., 2022)). However, given a linear model, it is natural to accept added stochasticity in the lack of nonlinearity (more on that in Section 5). On the right, we picked the best configuration ($T = 65$ in this case) to approximate the natural distribution. Notice, that the final generated distribution depends on the choice of parameters, where one can control the mean of the generated spectrum (this might be a feature for some applications, such as segmentation via diffusion, etc.). It might be interesting to derive the optimal parametrization for the convergence of the linear model - we leave this for future work. In addition to the convergence in mean, we included the standard deviation of the natural and generated samples, resulting in a decent fit to the target distribution.

The form of projection in Equation 32 can also help illustrate the appearance of higher frequencies during (stochastic) sampling. In Figure 7 we plot the mean distribution of the projection of generated samples over the clean principal components. The color is linked with the time - lighter colors as $t \to 0$. We can see that components with higher indices are apparent towards the end of the reverse process, indicating that higher frequencies appear later in the generation process.

We now turn to analyze the second sampling procedure, considering the loss defined in Equation 13 and repeated estimations of $x_0$. The generation starts from the denoising of $\xi_T$ by $D^*_{T \to 0}$ (defined in Equation 14), to obtain the first estimate of $x_0$, $D^*_{T \to 0} \xi_T$. The next denoiser is optimal considering the noise level $\bar{\sigma}_{T-1}$, so prior to its application, we add the next noise instance, $\xi_{T-1}$. Thus, the iteration in this denoising chain is given by

$$x_{t-1} = D^*_{t \to 0} x_t + \xi_{t-1}, \tag{33}$$

where again $\xi_t \sim \mathcal{N}(0, \bar{\sigma}_t^2 \mathbb{I})$. similarly to the former case, the final generated output $\hat{x}$ can be expressed as

$$\hat{x} = \Sigma_{t=0}^{T} \Pi_{\tau=0}^{t} D^*_{\tau \to 0} \xi_t = D^*_{0 \to 0} \cdots D^*_{T \to 0} \xi_T + \cdots + D^*_{0 \to 0} \xi_0. \tag{34}$$

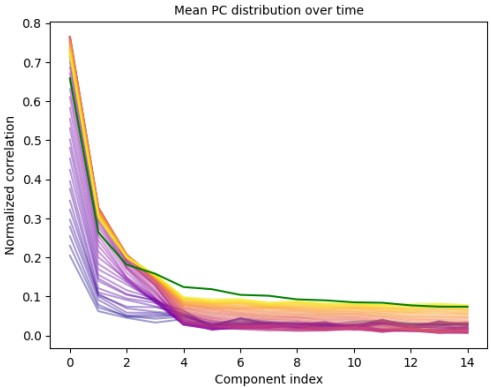

Figure 7: The empirical distribution of generated images over time - lighter colors for larger $t$. The high indices (i.e., high frequencies) appear at the later stages of sampling.

The difference between the generation path in Equation 34 and the one described in Equation 20 is in the applied denoisers, where the former utilizes the denoiser defined in Equation 14, and the latter employs the finite sample PCA denoiser (defined in Equation 18). In addition, the accompanying noise schedules should match the denoiser: $\{\sigma_t\}$ for the PCA denoiser and $\{\bar{\sigma}_t\}$ considering Equation 14.

Notice, that in this case as well, if we inspect the first element in Equation 34, i.e., $D^*_{0\to0}\cdots D^*_{T\to0}\xi_T$, the dominant direction is concentrated in the first eigenvector of $\Sigma_0$. This can be seen by looking at the diagonalization of $D^*_{t\to0}$,

$$D^*_{t\to0} = (\Sigma_0 + \bar{\sigma}_t^2\mathbb{I})^{-1}\Sigma_0 \tag{35}$$

$$= U_0 \begin{pmatrix} \frac{\lambda_0}{\lambda_0+\bar{\sigma}_t^2} & & \\ & \ddots & \\ & & \frac{\lambda_{r-1}}{\lambda_{r-1}+\bar{\sigma}_t^2} \end{pmatrix} U_0^\dagger,$$

since $\frac{x}{x+a}$ is monotonically increasing for $x, a \geq 0$. Thus, similarly to the case described in Equation 20, the generated output can be interpreted as a sum of high noise levels that were repeatedly correlated to estimate the leading data eigenvector, and lower noise levels that sample the entire data spectrum, in accordance with our discussion in Section 4.1.

## 5 Empirical Extension to Deep Denoisers

In the linear case described above, the optimal denoiser is given by PCA projections. These denoisers are computed with the training data, and their principal components do not depend on the input noise in the reverse process. When the denoiser is nonlinear, and might be implemented using a deep neural network, its input-output mapping can be locally expressed via the network Jacobian estimated at the inference point, $\nabla D_{net}(x_t)$, by

$$D_{net}(x_t) = \nabla D_{net}(x_t)x_t = V_t\Lambda_t V_t^\dagger x_t, \tag{36}$$

where $V_t\Lambda_t V_t^\dagger$ denotes the eigen decomposition of the Jacobian calculated at $x_t$. For simplicity, we assume that the Jacobian is symmetric and non-negative (which is approximately true (Mohan et al., 2020)). Note that in this case, the denoising base depends also on the input image (on top of the noise level). Although the network is non-linear, we can follow the generation path in the sampling process and inspect the basis of the network Jacobians calculated at the intermediate sampled points $x_t$. We can then trace $\sin\theta_J = \sqrt{1 - \langle v_i^t, v_i^{t=0}\rangle^2}$ where the subscript "J" stands for Jacobian, $v_i^t$ is the $i^{th}$ column in $V_t$ defined in Equation 36, in a similar way to our simulations of the linear case (Figure 2). This can be calculated per generation path, where $x_0$ is the final generated image, and $V_0$ is the basis of the Jacobian calculated at this final point.

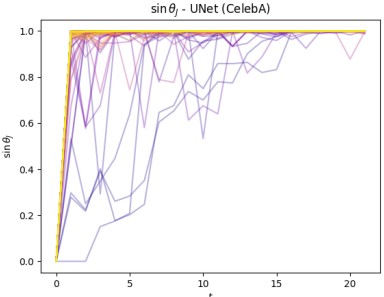 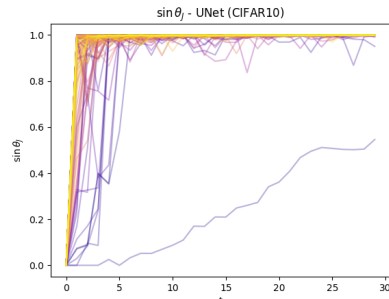

Figure 8: Image generation - the sine of the angle between Jacobian eigenvectors at the final generated image ($t = 0$) and intermediate iterations ($t > 0$). The diffusion model includes a UNet-based denoiser trained on CelebA (left) or CIFAR10 (rignt). Color by index (the darker the color the lower the index, referring to columns of the Jacobian basis $V_t$). The Jacobians of the nonlinear denoiser conform to the behavior of the linear model.

Figure 8 shows $\sin\theta_J$ calculated using the Jacobians of a UNet based diffusion model, described in Ning et al. (2023). This model was simply chosen as the [1]state-of-the-art in the task of image generation considering the CelebA dataset at the time of writing this paper. We used the default settings and calculated the Jacobians at the final iterations. We plot the results for the leading 300 Jacobian eigenvectors, where the color is assigned by the index - darker colors for lower indices $i$. We repeated the experiment sampling images from the CelebA dataset (left) and CIFAR 10 (right). Even though the denoising model is far from linear, the decay of the angle between the denoising basis in high noise levels and the natural denoising basis is similar to the decay in the linear case (compare to Figure 2). In this case as well, the correlation of the low indices (and hence low frequencies) withstands higher noise levels, thus appearing first in the generation process. As this is the basis of our analysis comparing the reverse diffusion process to power iteration, this experiment shows that our analysis is relevant in a broader context and not just in the simplified linear case. For more experiments considering other architectures, please see Appendix F.

This analysis focuses on the local behavior of the nonlinear denoiser at the end of the generation process, demonstrating its similarity to a linear denoising chain. Each plot represents a single generation path, not the overall distribution of generated outputs.

While linear diffusion models are easy to analyze, they may struggle to generate complex datasets. Nonlinear models, on the other hand, can navigate a diverse set of linearized regions during the generation process (as illustrated in Figure 8). This allows them to generate diverse data even without added noise, unlike linear models which ultimately converge in mean to a single point (Theorem 4.3) and therefore require noise injection for diverse outputs. This contrasts with some deterministic nonlinear samplers (e.g., (Lu et al., 2022)) that do not rely on added noise. The linearization approach we use is accurate at the sampled points in the generation trajectory, we cannot express the sampling procedure as strictly matrix multiplication (but rather as the composition of denoisers). However, if the denoiser is near linear in the sense that it has limited curvature around these sampling points, then our result might be approximately applied. We believe this is an interesting direction and leave its exploration for future work.

## 6 Conclusion

In this paper, we discuss a simple diffusion model with a linear denoiser and normalization free sampler, that allows us to cast the diffusion problem as noisy PCA, and make the connection to the spiked covariance model assuming that the natural data distribution resides in a low dimensional subspace. This enables us to show that in the linear case, the generation process acts as a "correlation machine", where initial random noise is repeatedly correlated to noisy estimations of the natural data basis, to finally embody the true distribution,

---

[1]https://paperswithcode.com/sota/image-generation-on-celeba-64x64

in a manner similar to the power iteration method. We show that in this process, low frequencies emerge earlier, and more data contributes to a richer representation per the same diffusion configuration. Finally, we demonstrate the relevance of our analysis also in a deep, non-linear diffusion denoiser.

We acknowledge the limitation of admitting a linear model, with its lack of ability to represent the complex data and often expected of diffusion models. In addition, studying whether a similar correlation-based structure persists in latent-space diffusion, where the manifold hypothesis may no longer apply, is an interesting direction for future work. While our theoretical setting is modest, we empirically demonstrate how our observations deduced from a simple linear model and classic theory (Johnstone, 2001; Nadler, 2008) are relevant to more general models and datasets. This enables us to shed light on the internal mechanism powering this technology, and connect it to a rich pool of theory and prevalent methods such as power iteration.

## Acknowledgments

The authors extend their sincere gratitude to Giulio Biroli, Gabriel Peyré, Mojtaba Ardakani, Tom Duerig, Tom Tirer and Shady Abu-Hussein for their valuable feedback. RG and DW were partially supported by the Israeli Innovation Authority, the Israeli council of higher education and the center for AI and Data Science at Tel University. RG and DW are also thankful for KLA and Google for their support.

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

# A  Power Iteration and its Convergence

Power Iteration is a simple algorithm used to compute the dominant eigenvalue and its corresponding eigenvector of a matrix. It iteratively refines an initial random vector by multiplying it by the matrix, which gradually aligns with the eigenvector corresponding to the largest eigenvalue. For a thorough introduction to the method, see, e.g. Andrilli & Hecker (2023). Given a square matrix $A \in \mathbb{R}^{n \times n}$, the goal is to compute the dominant eigenvalue $\lambda_1$ and its corresponding eigenvector $v_1$. The Power Iteration algorithm is defined as follows:

---
**Algorithm 1** Power Iteration Algorithm

---
**Input:** Matrix $A \in \mathbb{R}^{n \times n}$, initial vector $v_0 \in \mathbb{R}^n$, number of iterations $k$
**Output:** Approximate dominant eigenvector $v_k$
Normalize the initial vector: $v_0 \leftarrow \frac{v_0}{\|v_0\|}$
**for** each iteration $i = 1, 2, \ldots, k$ **do**
    $v_i \leftarrow A v_{i-1}$
    Normalize $v_i \leftarrow \frac{v_i}{\|v_i\|}$
**end for**
**return** $v_k$

---

The algorithm starts with an arbitrary vector $v_0$, which is normalized to ensure numerical stability. In each iteration, the vector $v_i$ is updated by multiplying it by the matrix $A$, followed by normalization. After $k$ iterations, the vector $v_k$ is expected to be close to the eigenvector corresponding to the largest eigenvalue of $A$.

## A.1  Convergence Analysis

Let $A$ be a square matrix with eigenvalues $\lambda_1, \lambda_2, \ldots, \lambda_n$, where the eigenvalues are ordered such that $|\lambda_1| > |\lambda_2| \geq \cdots \geq |\lambda_n|$. Denote the corresponding eigenvectors by $v_1, v_2, \ldots, v_n$, where $v_1$ is the eigenvector corresponding to the dominant eigenvalue $\lambda_1$.

The key idea behind Power Iteration is that, after sufficient iterations, the sequence of vectors $v_i$ converges to the eigenvector associated with $\lambda_1$, under certain conditions.

Let $v_0$ be the initial vector, which can be expressed as a linear combination of the eigenvectors of $A$:

$$v_0 = \sum_{i=1}^{n} \alpha_i v_i$$

where $\alpha_i$ are scalar coefficients. After applying the matrix $A$ in each iteration, we obtain the sequence of vectors:

$$v_i = A v_{i-1} = A \left( \sum_{i=1}^{n} \alpha_i v_i \right) = \sum_{i=1}^{n} \alpha_i \lambda_i^i v_i$$

Thus, the $i$-th iteration amplifies the component of $v_0$ along the direction of the eigenvector corresponding to the eigenvalue $\lambda_1$, while the other components decay at a rate proportional to the magnitude of their respective eigenvalues. As the iterations proceed, the contribution of the eigenvectors associated with smaller eigenvalues diminishes, and the vector $v_i$ becomes increasingly aligned with $v_1$, the eigenvector corresponding to $\lambda_1$.

Formally, we express the evolution of $v_i$ as:

$$v_i = \lambda_1^i \alpha_1 v_1 + \lambda_2^i \alpha_2 v_2 + \cdots + \lambda_n^i \alpha_n v_n$$

The relative influence of the eigenvectors corresponding to $\lambda_2, \lambda_3, \ldots, \lambda_n$ decays exponentially as $i \to \infty$ because $\lambda_1 > |\lambda_2| \geq \cdots \geq |\lambda_n|$. Specifically, the error in approximating $v_1$ decreases at a rate proportional to $\frac{|\lambda_2|}{|\lambda_1|}$, leading to the following convergence result:

$$\frac{\|v_i - \lambda_1^i v_1\|}{\|v_1\|} \leq C \left( \frac{|\lambda_2|}{|\lambda_1|} \right)^i$$

for some constant $C$, where $\|\cdot\|$ is the vector norm (usually the Euclidean norm).

Therefore, the Power Iteration algorithm converges to the dominant eigenvector $v_1$ at a rate determined by the ratio of the magnitudes of the first and second largest eigenvalues, $\rho = \frac{|\lambda_2|}{|\lambda_1|}$. If $\lambda_2$ is much smaller than $\lambda_1$, convergence is fast. However, if $\lambda_2$ is close to $\lambda_1$, convergence can be slow, requiring more iterations to achieve a satisfactory approximation. The convergence is linear, with the error decaying exponentially as the number of iterations increases. For a matrix $A$ with a well-separated dominant eigenvalue $\lambda_1$ (i.e., $|\lambda_1| \gg |\lambda_2|$), Power Iteration converges quickly, typically in $O(\log(\epsilon)/\log(\rho))$ iterations to achieve an error of size $\epsilon$.

## B    The Spiked Covariance Model

The spiked covariance model is a widely studied statistical framework that describes the behavior of the sample covariance matrix when a low-rank signal is embedded in high-dimensional noise. Originally introduced by Johnstone (2001), the model captures scenarios where a small number of principal components (the "spikes") are distinguishable from an otherwise isotropic Gaussian noise background.

Formally, let $X \in \mathbb{R}^{p \times n}$ be a data matrix with $n$ i.i.d. observations drawn from a multivariate normal distribution:

$$X_i \sim \mathcal{N}(0, \Sigma), \quad \Sigma = \sigma^2 I_p + \sum_{k=1}^{r} \theta_k v_k v_k^T,$$

where $\sigma^2 > 0$ is the noise variance, $r \ll p$ is the number of spikes, $\theta_k > 0$ are the spike strengths, and $v_k \in \mathbb{R}^p$ are orthonormal eigenvectors associated with the signal. The goal is to understand the spectral properties of the empirical covariance matrix $\hat{\Sigma} = \frac{1}{n} X X^T$, especially in the high-dimensional regime where $p, n \to \infty$ with $p/n \to \gamma \in (0, \infty)$.

A key insight of the spiked covariance model is that in the high-dimensional limit, the bulk of the spectrum of $\hat{\Sigma}$ follows the Marchenko–Pastur distribution (Marchenko & Pastur, 1967), while sufficiently strong spikes result in outlier eigenvalues. Baik et al. (2005) discovered a phase transition - now known as the BBP phase transition - characterizing when a spike separates from the bulk spectrum. Specifically, for a single spike $\theta$, the corresponding sample eigenvalue $\lambda$ detaches from the bulk if $\theta > \sigma^2 \sqrt{\gamma}$, and the associated eigenvector becomes asymptotically correlated with the true signal direction.

Subsequent work extended these results to multiple spikes (Paul, 2007), non-Gaussian settings (Benaych-Georges & Nadakuditi, 2011), and Bayesian and optimal shrinkage estimation (Donoho et al., 2018). Computationally, algorithms such as PCA and its variants remain the primary tool for extracting low-rank signals, although care must be taken in interpreting components near the detection threshold.

The spiked covariance model has profound implications in both theoretical and applied contexts. It provides a rigorous foundation for principal component analysis (PCA) in modern regimes where the dimension is comparable to or larger than the sample size. Applications span signal processing, wireless communications, genomics, and finance - anywhere that latent low-dimensional structures are inferred from noisy data.

Furthermore, the model serves as a testbed for understanding fundamental limits in statistical estimation, including the detectability of weak signals, the performance of eigenvalue shrinkage methods, and the reliability of inference under model misspecification.

## C  Frequency Characteristics of Principal Components

Principal Component Analysis (PCA) is a cornerstone technique in statistical data analysis and dimensionality reduction. Given a data matrix $X \in \mathbb{R}^{p \times n}$ (with each column representing an observation), PCA identifies an orthogonal basis of directions $u_1, u_2, \ldots, u_p$ (the principal components or PCs) such that the projection of the data onto each $u_i$ captures successively less variance.

When applied to natural images, PCA components often exhibit a hierarchical structure, with lower-index components capturing broad, low-frequency features, and higher-index components capturing finer, high-frequency details. This phenomenon is particularly evident when analyzing small image patches (e.g., $8 \times 8$ or $16 \times 16$ pixels) extracted from natural images. The first few principal components often correspond to smooth, low-frequency patterns, while the later components capture high-frequency structures such as edges and textures.

This pattern is consistent with the fact that PCA decomposes data using the eigenvectors of the sample covariance matrix $\hat{\Sigma} = \frac{1}{n} X X^T$. When $X$ arises from smooth processes (e.g., discretized Gaussian processes, Brownian motion, or autoregressive time series), the covariance matrix has a Toeplitz or smooth kernel structure. Its eigenvectors, under suitable conditions, converge to sinusoidal or Fourier-like modes (Grenander & Szegő, 1958; Jolliffe, 2002). This behavior can be theoretically justified under models where the data-generating process is smooth or governed by a differential operator. In natural images, the covariance matrix typically exhibits a smooth, spatially invariant structure. This translation-invariance implies that the eigenvectors of the covariance matrix, i.e., the principal components, tend to be sinusoidal in nature. As the component index increases, these sinusoidal modes become increasingly oscillatory, corresponding to higher spatial frequencies, often visualized as "Fourier-like" PCA modes.

This behavior has been observed in various studies. For instance, Hancock et al. (Hancock et al., 1992) found that the principal components of natural images resemble derivatives of Gaussian operators, similar to those found in the visual cortex and inferred from psychophysics. In addition, eigenvalues in facial recognition exhibit smooth lighting and shading variations in early PCs, and finer texture or edge-like patterns in later PCs (Sirovich & Kirby, 1987).

## D  PCA Optimality And Other Linear Denoising Chains

In the main text we discuss a gradual denoising chain, where noise is iteratively projected onto cleaner PCA bases (as defined in 18). In the following, we will clarify the sense in which PCA is optimal, and present another linear denoising scheme, which will help to frame the subject of this work.

The optimal linear denoiser at time t in the $\ell_2$ sense is the minimizer of the loss

$$\ell_{t+1 \to t} = \mathbb{E}_{x_t, w} \left\| D_{t+1 \to t}(x_t + \sigma_t w) - x_t \right\|_2^2, \tag{37}$$

where $w \sim \mathcal{N}(0, \mathbb{I})$. For the brevity of writing, we locally denote $D_{t+1 \to t}$ by $D_t$, until Eq. 41. This can be minimized by deriving the expected loss

$$
\begin{aligned}
\mathbb{E}_{x_t, w} \left\| D_t(x_t + \sigma_t w) - x_t \right\|_2^2 &= \mathbb{E}_{x_t, w}[x_t^\dagger D_t^\dagger D_t x_t - 2x_t^\dagger D_t x_t + \sigma_t^2 w_t^\dagger D_t^\dagger D_t w_t + x_t^\dagger x_t] \\
&= \mathbb{E}_{x_t, w} \operatorname{Tr}\left[ D_t^\dagger D_t x_t x_t^\dagger - 2D_t x_t x_t^\dagger + \sigma_t^2 D_t^\dagger D_t w_t w_t^\dagger + x_t x_t^\dagger \right] \\
&= \operatorname{Tr}\left[ D_t^\dagger D_t \Sigma_t - 2D_t \Sigma_t + \sigma_t^2 D_t^\dagger D_t + \Sigma_t \right],
\end{aligned}
\tag{38}
$$

where we have used the fact that $w_t$ has zero mean. To derive the optimal linear denoiser, we have

$$\frac{d\ell}{dD_t} = 2D_t \Sigma_t - 2\Sigma_t + 2\sigma_t^2 D_t = 0, \tag{39}$$

and so

$$D_t = (\Sigma_t + \sigma_t^2 \mathbb{I})^{-1} \Sigma_t. \tag{40}$$

Notice, that in the limit of diminishing $\sigma_t$,

$$D_t = U_t \begin{pmatrix} \frac{\lambda_0}{\lambda_0 + \sigma_t^2} & & \\ & \ddots & \\ & & \frac{\lambda_{r-1}}{\lambda_{r-1} + \sigma_t^2} \end{pmatrix} U_t^\dagger \underset{\sigma_t \to 0}{\to} U_t U_t^\dagger \triangleq D_{\text{PCA}}^t. \tag{41}$$

Alternatively, this can be seen as the minimizer when we average also on the input noise variance. In this work, we focus on the iterative application of $D_{t+1 \to t}$, and use the theory regarding noisy PCA (Nadler, 2008) to analyze the convergence properties of this chain.

Given a similar $\ell_2$ loss, one might suggest an alternative denoising chain, using multiple estimation of $x_0$. The corresponding loss is thus

$$\ell_{t \to 0} = \mathbb{E}_{x_t, w} \|D_{t \to 0}(x_0 + \bar{\sigma}_t w) - x_0\|_2^2, \tag{42}$$

where $\bar{\sigma}_t$ is the overall added noise (see Section 4). The adequate denoising chain in this case is the application of $D_{t \to 0}$ to estimate $x_0$, followed by the addition of noise with the appropriate variance $\bar{\sigma}_{t-1}^2$, before the iterative application of $D_{t-1}$. In this case, the optimal denoiser is given by

$$D_t = (\Sigma_0 + \bar{\sigma}_t^2 \mathbb{I})^{-1} \Sigma_0 \tag{43}$$

$$= U_0 \begin{pmatrix} \frac{\lambda_0}{\lambda_0 + \bar{\sigma}_t^2} & & \\ & \ddots & \\ & & \frac{\lambda_{r-1}}{\lambda_{r-1} + \bar{\sigma}_t^2} \end{pmatrix} U_0^\dagger.$$

In order to generate data, this denoiser is applied on a series of noises $w_t$, where $w_t \sim \mathcal{N}(0, \bar{\sigma}_t^2 \mathbb{I})$ for some schedule $\{\bar{\sigma}_t\}_{t=0}^T$. The generation starts from the denoising of $w_T$ by $D_{T \to 0}$, to obtain the first estimate of $x_0$, $D_{T]tto0} w_T$. The next denoiser is optimal considering the noise level $\bar{\sigma}_{T-1}$, so prior to its application, we add the next noise instance, $w_{T-1}$. Thus, the iteration in this denoising chain is given by

$$x_{t-1} = D_{t \to 0} x_t + w_{t-1}, \tag{44}$$

where again $w_t \sim \mathcal{N}(0, \bar{\sigma}_t^2 \mathbb{I})$. Due to the linearity of the denoisers, the final generated output $x_g$ can be expressed as

$$x_g = \Sigma_{t=0}^T \Pi_{\tau=0}^t D_{\tau \to 0} w_t = D_0 \cdots D_{T \to 0} w_T + \cdots + D_0 w_0. \tag{45}$$

Notice, that in this case as well, if we inspect the first element, i.e., $D_{0 \to 0} \cdots D_{T \to 0} w_T$, the dominant direction is concentrated in the first eigenvector of $\Sigma_0$ (since $\frac{x}{x+a}$ is monotonically increasing for $x, a \geq 0$). Thus, similarly to the case described above, the generated output can be interpreted as a sum of high noise levels that were repeatedly correlated to estimate the leading data eigenvector, and lower noise level that sample the entire data spectrum, in accordance with our discussion in Section 4.1.

## E  The Derivation of Perturbation Bounds from Nadler (2008)

In our analysis, we rely on nonasymptotic bounds on the angle between the empirical principal component vector and the true population component direction, as derived by Nadler (2008). These results pertain to the spiked covariance model in a finite-sample regime.

Consider the spiked covariance model:

$$x = uv + \sigma\xi,$$

where $u \in \mathbb{R}$ is a scalar latent variable with zero mean and unit variance, $v \in \mathbb{R}^p$ is a fixed signal direction, $\xi \sim \mathcal{N}(0, I_p)$ is Gaussian noise. Given $n$ i.i.d. samples $\{x^{(\nu)}\}_{\nu=1}^n$, we compute the sample covariance matrix $S_n$, and denote its leading eigenvector as $v_{\text{PCA}}$. The angle between $v_{\text{PCA}}$ and the true direction $v/\|v\|$ is denoted by $\theta_{\text{PCA}}$, with

$$\sin\theta_{\text{PCA}} = \sqrt{1 - \langle v_{\text{PCA}}, v/\|v\| \rangle^2}.$$

Assuming $v = \|v\|e_1$ and defining the signal strength $\kappa = \|v\|s_u$, where $s_u^2 = \frac{1}{n}\sum_{\nu=1}^n (u^{(\nu)})^2$, Nadler provides a high-probability upper bound:

$$\sin\theta_{\text{PCA}} \leq \frac{\sigma}{\kappa}\sqrt{\frac{p-1}{n}}\left(1 + \frac{2\sigma s_1}{\kappa\sqrt{n}}\right)\left(1 + \frac{s_2}{\sqrt{p-1}}\right) + \frac{4\sqrt{2}\sigma^2}{\kappa^2}\cdot\frac{p}{n}\cdot\frac{1}{1 - \left(\frac{2\sigma s_1}{\kappa\sqrt{n}}\right) - \left(\frac{\sigma^2}{\kappa^2}\right)},$$

which holds with probability at least $1 - \varepsilon_1 - \varepsilon_2 - \varepsilon_3 - \varepsilon$, where $s_1, s_2$ are parameters controlling the tail probabilities of the underlying Gaussian and chi-squared distributions (Nadler, 2008, Eq. (2.13)), and

$$\varepsilon = \exp\left(-\frac{p}{2(\sqrt{5}+2)^2}\right), \quad \varepsilon_1 = \Pr\{|N(0,1)| > s_1\},$$

$$\varepsilon_2 = \Pr\left\{\left|\frac{\chi_{p-1}^2}{p-1} - 1\right| > \frac{s_2}{\sqrt{p-1}}\right\}, \quad \varepsilon_3 = \Pr\{\chi_1^2 > s_3\}.$$

This bound shows that the misalignment of the empirical principal direction with the true signal direction scales approximately as

$$\sin\theta_{\text{PCA}} \lesssim \frac{\sigma}{\|v\|}\sqrt{\frac{p}{n}}, = \frac{\bar{\sigma}_t}{\lambda}\sqrt{\frac{d}{n}},$$

with additional correction terms involving higher-order noise and concentration effects, where the r.h.s. is the equivalent notation in the main text. In addition, Nadler (2008) show in their Corollary 1 that this bound is sharp, in the sense that it is also the the expected value for $\sin\theta_{PCA}$. This relationship justifies the dependence on $\bar{sigma}, \lambda, n, d$ in our own estimate and supports the analysis used in Section 4 of this paper.

## F  Additional Non-linear Diffusion Model Simulations

In this section we provide additional experiments to support Section 5. We follow the generation path in the sampling process of a single image and inspect the basis of the network Jacobians calculated at the intermediate sampled points $x_t$. We then trace $\sin\theta_J = \sqrt{1 - \langle v_i^t, v_i^{t=0}\rangle^2}$ where the subscript "J" stands for Jacobian, $v_i^t$ is the $i^{th}$ column in the Jacobian diagonalizing basis $V_t$ defined in Equation 36, in a similar way to our simulations of the linear case (Figure 2). Each plot represents the generation path of a single image, where the color is by index - the darker the color, the lower the index.

We reproduce the experiment held in Section 5 using two popular implementations (Karras et al., 2022) of two different model architectures: "DDPM++ cont. (VP)" and "NCSN++ cont. (VE)" models by Song et al. (2021c), with the default parameters and 36 generation iterations. We repeat the experiment in conditional cifar-10 models, reported in Figure 9. In these cases as well, we see that lower indices preserve correlation to the generated image in higher noise levels, towards the end of the generation process.

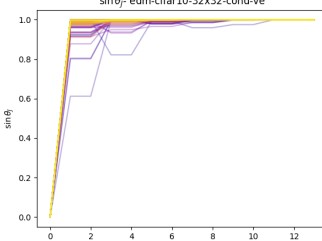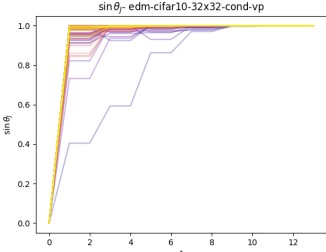

Figure 9: Image generation - the sine of the angle between Jacobian eigenvectors at the final generated image ($t = 0$) and intermediate iterations ($t > 0$), for "DDPM++ cont. (VP)" (right) and "NCSN++ cont. (VE)" (left) models. Color by index (the darker the color the lower the index, referring to columns of the Jacobian basis $V_t$). The Jacobians of the nonlinear denoiser conform to the behavior of the linear model.

