# OpenReview forum: "The Diffusion Process as a Correlation Machine: Linear Denoising Insights"
_TMLR — Accepted by TMLR_

### Review · Reviewer_6Q5P · 2025-04-16

**Summary Of Contributions:**

The authors analyze the generation process, considering a linear model as the score estimator or denoiser. In this case, the optimal denoiser is $D_t=(\Sigma + \sigma^2 I)^{-1} \Sigma$, where $\Sigma$ represents the data covariance and $\sigma$ is the noise level.

Consequently, the diffusion generative process simplifies to the sum of noise projected by the trained linear denoiser: $D_0 \cdots D_T \xi_T + \cdots + D_0 \xi_0.$

This leads to a generalization of the power iteration method.
- The first part $D_0 \cdots D_T \xi_T$ aligned with the first eigenvector.
- The last part $D_0 \xi_0$ closely resembles the clean data.

**Audience:**

Yes

**Broader Impact Concerns:**

None.

**Claims And Evidence:**

No

**Requested Changes:**

Explain following related material in this paper.
- correlation machine
- spiked covariance model
- low frequency feature in terms of linear space

Section 3 of "Problem Setup" is confusing.
- Begin with clear definitions of the practical setups in literature and theoretical ones used in this paper, then explain the differences between them.
- Differentiate the definitions and derived properties.
- $D_t$ has different meanings in equations (7), (9), and (13); use distinct notation.
- $\Sigma_t$ in equation (9) is undefined before.
- Define $\lambda_i$ and indicate $t$ in the notation since $\lambda_i$ depend on $t$.
- Clarify the definition of $X_t$ in equation (11).
- What $\bar{\sigma}$ is and why both $\sigma$ and $\bar{\sigma}$ are used.

Section 4 is incomplete and lacks explanation.
- Equation (15) is incorrect. Replace $U_t$ with the approximated of basis $\hat{U}_t$ as noise can also perturb eigenvectors along with eigenvalues.
- What does assumption, eq.(17) imply?

**Strengths And Weaknesses:**

This paper presents an intriguing topic by theoretically analyzing the well-known phenomenon in diffusion models where coarse, low-frequency features are generated first, followed by finer, high-frequency features.

The linearization strategy and power method approach offer valuable insights without being overly simplistic.

However, the conclusions drawn in the last three paragraphs on page 9 (The first eigenvector, The entire spectrum, and In between) are weak. To make the paper more compelling, the authors could explore additional findings, such as how the frequency of features generated varies with t, the differences between stochastic and deterministic samplers in this context, and the impact of varying the number of training samples on t.

Moreover, the paper is not easy to follow because of its presentation style. (See below)

---

> ### Author Response · Authors · 2025-05-09
>
> Thank you for your detailed review. We provide below answers to all points raised.
>
> **Improving structure.** We have rearranged Section 3 (and 4) according to your comments, added the requested materials, and clarified all queried implications in the main text. We will update the manuscript file once all reviews are in, as advised by TMLR's editorial policies. We now provide a detailed per point answer to your concerns. Specifically, we provide below details about some of the points raised:
>
> **Additional findings.**
> Some of the additional findings you suggested are already mentioned in the paper - we have emphasized them and added to their discussions in the main text.
> * How the frequency of features generated varies with t - We have added a figure illustrating this point. In Figure 7 (in the revised manuscript), we plot the mean distribution of the projection of generated samples over the clean principal components. The color is linked with the time - lighter colors for larger t. It shows that the components with higher indices (i.e., higher frequency) are apparent in later times, indicating that higher frequencies appear later in the generation process.
>  * Stochastic and deterministic samplers - The sample trajectories we consider in the paper are stochastic. However, our analysis also covers a deterministic sampling path, where the denoisers are repeatedly applied, with no added noise. This deterministic sampling is in fact portrayed by the first summand in the generated output (Eq. 18, or Eq. 20 in the revised manuscript), which we show to converge in mean to the first eigenvector, similarly to the power iteration. This also implies that this deterministic sampling is insufficient to capture the training data in the linear case. We have added this discussion to the main text.
>   * Varying the number of training samples on t - The result in Equation 17 (Eq. 19 in the revised version) shows the dependence on the number of training samples, in the rank-1 case. In the context of linear diffusion, we empirically show that the behavior captured by Eq. 17 is apparent also in higher ranks (Figure 3). We interpret these results as the emergence of higher frequency components at higher noise levels, thereby capturing more nuances in the generated data.
>
> **Related material:**
> * "Correlation machine" - we use this term as a figure of speech, to describe a system that can be characterized by its influence on the input - output correlation. We have added clarifications to the main text.
>  * Spiked covariance model - We have added an appendix with the relevant background regarding the spiked covariance model.
>  * low frequency feature in terms of linear space - We have added an appendix with the needed clarifications and relevant background.
>
> **"Problem Setup"**:
> We have rearranged this section, and separated the standard definitions and our linear problem setup.
>
> Section 3:
>    * We used distinct definitions of $D_t$.
>    * We defined $\Sigma_t$ before Eq. (9).
>    * We defined $\lambda_i$ and added a clarification regarding it being time dependent. However, we chose to omit an explicit notation for brevity.
>    * We clarified the definition of $X_t$ in Equation (11).
>    * $\sigma$ is the intermediate noise between $t\to t + 1$; $\bar{\sigma}$ is the overall noise variance, going from $0\to t$. We have emphasized this and added a clarification to the main text.
>
> Section 4:
>    * Regarding Eq. 15 (Eq. 17 in the revised manuscript) - You are correct to notice that the noise also causes a perturbation of the eigenvectors - it was noted with the subscript $t$. However, our former notation "hides" the perturbation caused by the finite sample approximation (it is not the main focus of our paper, as we explain in the paragraph following Eq. (15)). To improve our readability and rigor, we redefined the noisy PCA basis (now in Eq. 17) by $\hat{U}_t$, as requested, and added clarifications regarding this point throughout the paper.
>    * Eq. 17 is not an assumption, but a proven result presented by Nadler in [34]. They study the sample complexity of the rank-1 case, and bound the angle between the finite sample and population PCA under a spiked covariance model similar to ours (only its rank-1, and ours is rank-r). Their result implies linear sample complexity and shows that the leading eigenvector rotates at a rate proportional to the noise level. Our assumptions 4.1, 4.2 are expansions of the result in (17) to the rank-r case - Assumption 4.1 describes the correlations of the same index with its noisy versions, and Assumption 4.2 describes the cross index correlation (that are relevant due to the noise). We have added this clarification to the main text.

---

### Review · Reviewer_3BeG · 2025-05-21

**Summary Of Contributions:**

This paper considers diffusion models in a setting where the reverse process is driven by a linear (learned) drift function - and attempts at bringing some understanding to general behavior of diffusion models by analyzing/demonstrating the properties of the learned score.

**Audience:**

Yes

**Broader Impact Concerns:**

no concerns

**Claims And Evidence:**

Yes

**Requested Changes:**

1) References are poorly formatted. At the moment, equations and references uses the same formatting. E.g. when the authors mention (19), it is unclear if the equation is referred to or a citation.

2) What is $\Sigma_t$ in eq. (9), please clarify.

3) While I appreciate the full derivation is in Appendix, lines following eq. (9) and (10) are quite unclear. First $D_t$ is given as the expression in (9) without clarifying $\Sigma_t$, then in the limit $\sigma_t \to 0$, some expression is given as the PCA denoiser.

- What does the PCA denoiser mean?
- What do you mean by "the cleaner version of the training dataset"?

please clarify.

4) Fig. 1 caption would use some clarification. The first line is the denoising process, the second line is the target image? please clarify.

5) Could you make the derivation of the approximation in eq. (17) self-contained please - moving from [34] the relevant derivation?

6) In general, the results could be stated better and notation is confusing. I will compile some examples here, but this generalize and makes things unreadable.

For example, $x_{0:T}$ is used for diffusion model trajectory. But the training set is denoted $X = \{x_1, \ldots, x_n\}$! Further more, before eq. (11), $X_{t-1}$ notation is introduced, "cleaner version" (as referred above), left unclear. $\bar{\sigma}_t$ is not defined when it is first used in (13), deferred, making things unclear. in Theorem 4.3, still define $\hat{x}_T$ to make the statement self-contained. Please give a proofread to the paper.

7) How are your results impacted by several approximations you are making? Why are they okay, how are they justified? Please add a remark about these.

8) Can anything be said in Section 5, that is more interesting for general denoisers? If a denoiser "near-linear", can most of your results be applied using Jacobians?



"Assuming 4.2, 4.1 -> Assuming 4.1, 4.2"

**Strengths And Weaknesses:**

Strengths:

- it is an interesting idea to look at the spectral structure of linear denoisers. This is also insightful to understand the emergence of data structures during the generation process (e.g. early low frequency elements), which can lead to insightful developments.

Weaknesses:

- I think the paper has more weaknesses than the strengths. Overall I found the explanations confusing and complex, despite the paper essentially is in the "linear setting". There are multiple typesetting issues (pointed out below in requested changes) around referencing. Also I think figure captions do not read well, as well as the general message in the abstract. The paper would benefit lots from clarification both in text and explanations (see my requested changes).

---

> ### Author Response · Authors · 2025-06-07
>
> Thank you for your detailed review and suggestions. We have rearranged Section 3 and 4 and clarified all queried implications in the main text. We now provide a detailed per point answer to your concerns:
>
>    * We fixed the reference style.
>
>    * $\Sigma_t$ is the noisy covariance, at time $t$ (defined in Eq. 9 in the manuscript, originally in Eq. 14).
>
>    * We have rearranged Sections 3 and 4 for a more streamlined reading experience, the changes are marked in color in the revised manuscript. We now carefully define each of the denoisers, after defining $\sigma_t$ (the added noise between two diffusion steps, Eq. 7), $\Sigma_t$ (the noisy covariance, Eq. 9), and introduce the PCA denoiser as the limit of $D_{t+1 \to t}$ where the intermediate added noise goes to zero (Eq. 12).
>
>      * PCA denoiser is the projection onto the PCA components. It is introduced in Eq. 12 and explained in the text following it.
>      * The cleaner version of the training dataset is $X_{t-1}$. We altered the text in the paper to be clearer.
>
>  * We further decimated Fig. 1 to include a single row, where the reverse process runs from left to right.
>
>  * We added Appendix ``The Derivation of Perturbation Bounds from Nadler 2008'' including the derivation of Eq. 17 (now Eq. 19) from [34] (Boaz Nadler. Finite sample approximation results for principal component analysis: A matrix perturbation approach.)
>
>   * As mentioned above, we have rearranged Sections 3 and 4 to improve the readability of the paper. In addition, we address all of your requested changes in the main text.
>
>   * The main approximation we use is in Nadler's result (Eq. 19). They derive a finite sample bound for $\sin \theta_{\text{PCA}}$, including all factors arising from the signal - noise interactions (similarly to our Eq. 17). This is an intricate bound, with many details (we now summarize the relevant parts in an additional appendix, per your request). We chose to only consider the leading part of this bound, because it captures the essence of the bound while significantly simplifying the writing, increasing the approachability of our paper. This approximation is justified since this bound is shown to be sharp in Corollary 1 in Nadler 2008, and empirically in our simulations ( specifically in Fig. 2). We added a remark about this point to the main text.
>
>   * This is an interesting point. The linearization approach we used is accurate at the sampled points in the generation trajectory. Thus, we cannot express the sampling procedure as strictly matrix multiplication (but rather as the composition of denoisers). However, if the denoiser is near-linear in the sense that it has limited curvature around these sampling points, then our result might be approximately applied. We agree that this is an interesting direction, and we added it to the discussion in Section 5, suggesting to explore it in future work.
>
>  * We fixed the reference order.

---

### Review · Reviewer_MRyv · 2025-05-24

**Summary Of Contributions:**

The authors propose a novel approach to interpreting and understanding diffusion models, albeit primarily in a simplified linear setting. The resulting analysis brings about a better understanding of denoising process in diffusion as one of projections onto principal directions of the data model. The paper also shows experimentation on some real-world data to correlate the observations — both by applying the linear model to MNIST data, and by analyzing the Jacobian through the same analysis on CelebA/CIFAR10.

**Audience:**

Yes

**Claims And Evidence:**

No

**Requested Changes:**

**Requested Changes**
- Some diversity of experimentation on the non-linear model would help solidify the paper’s claim that the insights generalize to realistic settings as well. The authors consider two datasets, but at least 2-3 architecture choices, or models trained using varying strategies also conforming to the proposed analysis would be strongly supporting of this work
- Adding discussions on the extension to latent spaces, or to non-$\ell\_2$ losses/models would be interesting

The rest of my concerns are relatively minor. In general I found the notations to sometimes get overloaded, causing the reading experience to get a tad bit jarring sometimes, but these are easily addressable concerns:
 - The notation is equation (9) felt confusing to me. Would’t it be better to differentiate between the function $D\_t(\cdot)$ in Eqn (8) and the the optimum $D\_t^*(\cdot)$ defined in Eqn (9).
 - Similarly in Sec. 3, the use of $q$ to denote the data distribution, with $q(x\_t | x\_{t-1})$ denoting the transition probability. Why not call the data model $q\_d$?
 - Lastly, in Eqn (24), $D(x) = \nabla D(x) x$ is confusing notation. It would help to differentiate the Denoiser $D$, and maybe the Jacobain of the neural-net as $\nabla D\_{Net}$ or some such notation.
-In Fig. 6, it would help to have the scale of the y-axis aligned, since the primary goal is to bring to attention the fact that the correlations saturate to higher value in the middle figure. This insight is visually lost because of the scaling of the axes. Let all three figures have Y-axis in [0,1]. Please add Y-axis labels to Fig. 7
 - It feels weird to not use grayscale to be plotting MNIST in Fig. 1, since that is the standard approach in the literature. Maybe the authors to fix their plotting function to not print MNIST figures on a heat-map color gradient. (Unless there is some particular reason to use it here that I might have missed).

**Strengths And Weaknesses:**

**Strengths**:
 - The analysis presented is rigorous and well founded in general. It is straightforward to follow and understand (but for a few minor issues, check Requested Changes)
 - The presented experimentation validates the observations derived.
 - I believe the analysis opens up avenues for further understanding diffusion models, and the results derived could be of value to the broader community. For example, it might be interesting to see if the learning of low frequencies first comes as a consequence of the 2-norm formulation of the losses, and if there could be similar results derived for other denoising (and consequently, different noise) models.

**Weakenesses**:
 - The experimental extension to non-linear models is good, but maybe not entirely comprehensive. While it does seem that the proposed insights translate to the Jacobian’s eigenvalue and eigenvector analysis, it might’ve been more convincing to try it out on atleast a few more model/architecture variants. For one, given Ning et al., [35] (that this paper considers), one could consider the follow-up Li et al., ICLR 2024. Furthermore, could there be some correlation between the analysis carried out in the proposed approach, and in these works that deal with exposure bias in diffusion models (wherein the main claim is that the diffusion process is “two-stage” — first in moving generally towards to the data distribution, and second, in latching onto a single image that gets generated). Of course, I understand the constrains of doing full-data PCA on larger-dimensional data, so the authors could restrict themselves to the CIFAR-10, CelebA-64 scale.
 - Tying in to the point above, though I don’t expect significant experimentation here (but it would be a plus), is on how this analysis fares when dealing with latent-space diffusion, wherein the manifold hypothesis of data no longer holds, and the low-to-high-frequency distribution does not exist in the sense that the paper considers. Might be an interesting direction to address in the future at the very least, though I would then hope that the authors discuss it in the future scope.

[1] Li et al., ALLEVIATING EXPOSURE BIAS IN DIFFUSION MODELS THROUGH SAMPLING WITH SHIFTED TIME STEPS, ICLR 2024

---

> ### Author Response · Authors · 2025-06-07
>
> Thank you for your detailed review and enlightening suggestion. We now provide a detailed per point answer to your requested changes and suggestions:
>
> * Diversity of experiments - we have added more experiments with varying architectures, sampling methods and training configurations in Appendix "Additional Non-linear Diffusion Model Simulations". The additional results support our claim, and further demonstrate its applicability to the commonly used diffusion models.
>
> * The discussion regarding latent diffusion models and the connection to exposure bias are very interesting; thank you for suggesting them.
>     We added the context of exposure bias and the recent work by Li et al. (ICLR 2024) to Section 4. Their main insight that the diffusion process exhibits a “two-stage” behavior, initially guiding samples toward the data manifold before committing to specific modes is particularly relevant to our analysis.
>     While our work focuses on the reverse process in the linear setting, we indeed observe a similar two-phase dynamic: earlier time steps reconstruct low-frequency structure (aligned with leading eigenvectors), while later steps refine high-frequency details. This aligns with the interpretation of “latching onto a single image” in the second stage of generation as described by Li et al. Moreover, our PCA-based framework analytically supports this transition via eigenvalue-dependent correlation emergence, offering a complementary perspective.
>
>     Additionally, we agree that future extensions to latent-space diffusion, where the frequency decomposition may no longer hold under the manifold hypothesis, could be a valuable direction. We have noted this in the conclusion as an avenue for future work, particularly in settings where the learned representation may not be easily interpretable via PCA.
>
> * We rearranged Sections 3 and 4 to allow for a better reading experience, while introducing some clearer notations, including your suggestion to distinguish the optimal denoiser from the objective one.
>
> * We have noted the natural distribution by $q_D$, as suggested.
>
> * We added a clarification regarding the network Jacobian, and changed the network notation to $D_{net}$.
>
> * We have aligned the y-axis in Fig.6, added one to Fig. 7, and changed the MNIST plots to grayscale.

---

### Decision · Action_Editor_gbxg · 2025-10-02

**Recommendation:** Accept with minor revision

**Additional Comments:**

One of the reviewers points out that the (over)simplified or strong assumptions can cause conclusions that might be misleading for future research on the topic. Thus, my recommendation is to acknowledge this in the following parts of the paper and discuss possible limitations of the simplifying assumptions. These changes (acknowledging the concern by the reviewer in the paper) are minor. The comments by the reviewer are copied below verbatim.

>In particular, I believe the following conclusion and intermediate claims contribute to the incorrect conclusion.
>* Issues with the Conclusion in Section 4 (Page 10)
>
>The statement: "This also implies that deterministic sampling is insufficient to capture the training data in the linear case" contradicts existing results, as most sampling is conducted using discrete integrators. This contradiction raises doubts about whether the theoretical assumptions are oversimplified.
>
>* Problematic Approximations
>Figures 3 and 5 are presented as empirical support for the theory, but they may not be sufficient.
>1. Regarding Assumption 4.1 and Figure 3: Since $\sin \theta $ approaches 1 during the early phase of the diffusion process, the claim becomes less relevant.
>2. Regarding Assumption 4.2 and Figure 5: The figure does not clearly show that $U_iU_j^*$
resembles the identity matrix, which undermines the claim.

**Audience:**

Yes

**Audience Explanation:**

This work falls under topics that are interesting to the TMLR audience. The approach in the paper is insightful and interesting for the community, and it should at least interest those who have worked with diffusion models more closely.

**Claims And Evidence:**

Yes

**Claims Explanation:**

This paper takes a simplifying view to analyse properties of diffusion models, which sheds light on internal correlations within the system. As one of the reviewers said, the paper makes connections between existing, traditional machine learning methods and diffusion models. This view is meaningful, timely, and interesting for the community.